# Variations of Urban Thermal Risk with Local Climate Zones

**DOI:** 10.3390/ijerph20043283

**Published:** 2023-02-13

**Authors:** Jiaxing Xin, Jun Yang, Yipeng Jiang, Zhipeng Shi, Cui Jin, Xiangming Xiao, Jianhong (Cecilia) Xia, Ruxin Yang

**Affiliations:** 1Human Settlements Research Center, Liaoning Normal University, Dalian 116029, China; 2Jangho Architecture College, Northeastern University, Shenyang 110016, China; 3School of Marine Law and Humanities, Dalian Ocean University, Dalian 116023, China; 4Department of Microbiology and Plant Biology, Center for Earth Observation and Modeling, University of Oklahoma, Norman, OK 73019, USA; 5School of Earth and Planetary Sciences (EPS), Curtin University, Perth, WA 6845, Australia

**Keywords:** local climate zones, land surface temperature, single-window algorithm, heat risk index, Shenyang

## Abstract

Due to the differences in land cover and natural surroundings within cities, residents in various regions face different thermal risks. Therefore, this study combined multi-source data to analyze the relationship between urban heat risk and local climate zones (LCZ). We found that in downtown Shenyang, the building-type LCZ was mainly found in urban centers, while the natural- type LCZ was mainly found in suburbs. Heat risk was highest in urban centers, gradually decreasing along the suburban direction. The thermal risk indices of the building-type LCZs were significantly higher than those of the natural types. Among the building types of LCZs, LCZ 8 (open middle high-rise) had the highest average thermal risk index (0.48), followed by LCZ 3 (0.46). Among the natural types of LCZs, LCZ E (bare rock and paved) and LCZ F (bare soil and sand) had the highest thermal risk indices, reaching 0.31 and 0.29, respectively. This study evaluated the thermal risk of the Shenyang central urban area from the perspective of LCZs and combined it with high-resolution remote sensing data to provide a reference for thermal risk mitigation in future urban planning.

## 1. Introduction

Widespread urbanization and urban population increase have significantly changed the urban thermal environment and produced a severe urban heat island (UHI) effect [1]. This phenomenon describes the higher temperatures in urban areas than in the surrounding rural areas. Numerous research studies report that the UHI effect may lead to major problems such as decrease of air quality, increase of energy consumption, and change of vegetation phenology [2,3,4]; moreover, the UHI effect may be harmful to humans. Increased risks to mental health and well-being are associated with climate-sensitive health outcomes and systems caused by climate change [5,6]. An increased frequency of extreme heat events exacerbates health risks associated with cardiovascular disease, impairs agricultural productivity, and increases food insecurity in low-income areas [7,8,9]. Therefore, in the backdrop of frequent extreme heat events, studying the UHI effect and its associated thermal risks is necessary [10,11].

Due to differences in the development status of cities, the UHI effect and thermal risk status within cities differ. To better reflect the differences in heat island effects under different land surfaces, Stewart and Oke [12] proposed a regime of local climate zones (LCZs) that quantifies differences in land cover, composition, materials, and human activity to classify urban sub surfaces into 17 categories. The LCZ system has greatly promoted a comparative study of global UHI [13,14,15]. Regular and irregular grids are the main methods of LCZ mapping, which must be realized by quantifying the corresponding indicators. For example, the World Urban Database and access portal tools are used to construct training samples for supervised classification and classify remote sensing images with regular grids to achieve LCZ mapping [16]. The irregular grid approach usually requires detailed building plan data [17,18,19]. With advances in remote sensing technology, UHI studies from the perspective of LCZs have predominantly focused on surface UHIs. The research scale involves global, regional, and urban agglomerations and individual cities [20,21,22]; for example, Yang et al. [23] explored the surface temperature of various sized cities based on LCZs, using the Pearl River Delta urban agglomeration as the study area.

The 2021 World Risk Index reveals that China has a hazard index of 5.87%, an exposure of 14.29%, and a thermal vulnerability of 41.08% [24]. Early studies on heat vulnerability and exposure emphasized the role of socio-demographic factors, such as age, sex, and education, but this approach often failed to reflect the spatial heterogeneity of the population distribution across the entire range [25,26,27]. In addition, others have evaluated thermal vulnerability by calculating physiologically equivalent temperatures or thermal comfort indices [28,29,30]. With the diversity of data acquisition, biological variables such as land surface temperature (LST), land cover, and environmental factors were included in these studies [31,32]. Combining LST data simultaneously can improve the understanding of changes in the risk of extreme heat within cities, whereas land cover and environmental factors can better reflect people’s living environments [33]. Therefore, many scholars have conducted research on thermal risk based on the three components of the Clayton Triangle, namely, thermal hazard, exposure, and vulnerability, and different studies have adopted several indicators to quantify the three components [34,35]. For example, the number of days of extreme high temperatures and LST were used to represent the construction of the thermal hazard index, population density data were used to measure the thermal exposure index, and the elderly population combined with environmental factors were used to represent the thermal vulnerability index. Finally, the overall thermal risk index was calculated using weighted or unweighted methods to assess the thermal risk in the study area [36,37]. Therefore, based on the framework of the Clayton Triangle’s three components, this study used LST, population density, and people vulnerable to heat to represent heat hazard, exposure, and vulnerability, respectively, and then obtained the thermal risk index.

Some scholars have considered environmental factors while constructing a thermal risk index [38], and differences in heat health risk between urban, suburban, and rural areas have been analyzed [39]. Dong et al. [38] assessed the thermal risk under the Beijing heat island effect and found that impervious water was highly correlated with thermal health risk. Upon studying the thermal risk of Chongqing, Zhang et al. [40] combined vegetation, water bodies, and slopes to build a thermal vulnerability index, demonstrating consideration of differences in human bodies and the surrounding environment. In addition, the vector model has been used more frequently than the grid model to assess thermal risk. For example, Chen et al. [41] considered night light, vegetation index, digital elevation model, and other factors to evaluate the thermal risk of the Yangtze River Delta on a 250 m × 250 m grid but not the differences in the impact of land cover on thermal risk within cities. Jiang et al. [42] evaluated the thermal risk of Los Angeles on a 1 km scale by dimensionality reduction of the GEOS LST data. In addition, owing to the limitation of the spatial resolution of multi-source data, the evaluation of the thermal risk index cannot determine an optimal spatial scale. Most research on thermal risk is combined with statistical data, and the scale of statistical data is limited to the administrative scope, which often requires spatial quantification using an interpolation method [36]. Therefore, combined with existing multi-source data, this study investigates the thermal risk distribution from a scale of 100 m.

This study used the downtown area of Shenyang as the research object and applied the LCZ system to evaluate its thermal risk under different LCZ types and determine the distribution law of thermal risk among LCZs. The process consisted of three parts: (1) a grid of 100 was constructed, LCZ was divided, and LCZ mapping was conducted for the central urban area of Shenyang; (2) LST, population density, and statistical yearbook data were combined to construct the thermal risk index and conduct spatial visualization; and (3) spatial superposition of the two was conducted to analyze the thermal risk differences under different LCZs in Shenyang. Overall, the study can provide a reference for mitigating the heat island effect, plan reasonable urban land use, and improve urban livability.

## 2. Materials and Methods

### 2.1. Study Area

Shenyang is located at 41°48′ N, 123°25′ E in the southern part of northeast China and in the central part of Liaoning Province (Figure 1). It is a central city in northeast China approved by the State Council. In 2021, the total area of Shenyang was approximately 12,860 km^2^, with a permanent population of 9,118,000. In 2022, the People’s Government of Shenyang City issued the Action Plan for Building Shenyang into a National Central City to support Shenyang in accelerating the construction of a national central city and effectively enhancing its comprehensive strength and regional influence. This study selected Tiexi, Yuhong, Huanggu, Hunnan, Shenhe, Dadong, Heping, Shenbei New, and Sujiatun districts as the research areas. The demographic data of 2020 shows that 79% of the population of Shenyang is living in the central urban area; therefore, local residents are more vulnerable to the threat of high temperature.

### 2.2. Data

The data used for the study are shown in Table 1. LST inversion was conducted using Landsat data, and the entire process was realized using ENVI 5.3 software. The thermal risk index was calculated by combining population density and demographic data. The building-type LCZs were divided by building vector data, whereas the natural-type LCZs were divided by combining land use data. Sobrino et al. [43] demonstrated that a resolution of 100 m can appropriately reflect the LST differences between communities. At the same time, in order to match the world population density data, this study resampled the LST, and then analyzed it based on a 100-m grid. The whole research process is shown in Figure 2.

### 2.3. Methods

#### 2.3.1. LCZ Classification

Stewart and Oke [11] divide the entire LCZ system into architectural and natural LCZs. Therefore, combining with the natural conditions of the downtown area of Shenyang, relevant indices were calculated to divide the building-type LCZ, and the calculation formula is shown in Table 2. In addition, the classification of natural types was carried out based on land use data, and the whole process was realized in Arcgis10.5. Finally, the two were combined to obtain the final LCZ drawing (Table 3), in which the open type represents building density greater than 0.4, and the compact type represents building density less than 0.4.

#### 2.3.2. Calculation of Heat Risk Index

Different indicators have been used to quantify thermal risk indices, but they are all based on the framework of the Clayton Triangle. In this framework, thermal hazard, exposure, and vulnerability together constitute the thermal risk index. In this study, LST, population density, and susceptible population were respectively used to represent heat hazard, exposure, and vulnerability. However, existing studies do not have the most appropriate weight to construct the thermal risk index [39]. Therefore, the three indices of thermal hazard, thermal vulnerability, and thermal exposure are assigned the same weight in this study so as to obtain the final thermal risk index of the study area.

##### Heat Hazard Index

In terms of spatial and temporal resolution, the use of satellite remote sensing data has more advantages than the weather station interpolation method describing heat loss. Therefore, in this study, Landsat data was selected, and a single window algorithm was used to invert the LST and quantify the thermal hazard index by referring to relevant literature. Figure 3 shows the final land surface temperature inversion results

##### Heat Vulnerability Index

Many studies have shown that the elderly and children are the most vulnerable to heat owing to their special physiological conditions, resulting in a low tolerance to heat [44]. In addition, the Intergovernmental Panel on Climate Change (IPCC) report on thermal vulnerability was based on sensitivity and capacity. Considering the existing literature combined with the availability of data and consistency of spatial resolution, this study expressed the sensitivity by counting the population vulnerable to heat threats (children aged <15 years and elderly aged >65 years) and mapped it into the grid with the population density data (Figure 4).

##### Heat Exposure Index

The IPCC interpretation of heat exposure describes populations that may be adversely affected by high temperature. In combination with existing studies, this study measured heat exposure using population density as an indicator, quantified it using WorldPop gridded population density data, and used regression between population density data and demographic data to verify its accuracy. Increasing population density assumes that the heat exposure index increases from 0 to 1. See Figure 5 and Figure 6 for comprehensive spatial distribution and accuracy verification.

Finally, the three indices were normalized. As no standard weight was established, the same weight was assigned to the three indices, and then the final thermal risk index was calculated.

## 3. Results

### 3.1. LCZ Classification

Figure 7 shows the LCZ classification results for the entire study area. In terms of spatial distribution, building-type LCZs are principally located in the city center and along both banks of the Hun River, which is consistent with urban spatial planning. However, significantly more buildings were on the northern side of the Hun River than on the southern side, indicating a large space for development in realizing the coordinated and interactive development of the two sides of the urban inland river. For natural-type LCZs, LCZ A was distributed in the southeast of the study area, in relation to the ecological source planning of the southeastern hilly area of Shenyang, while the northern part of the study area was mainly composed of LCZ C. In terms of the number of building types (Figure 8), LCZ 10 had the highest proportion (15.23%), indicating that the study area was mainly dominated by open low-rise buildings, while LCZ 2 had the lowest proportion (0.09%). Among the natural types of LCZs, LCZ C was the largest (39.09%), followed by LCZ A (26.10%).

### 3.2. Heat Risk Index

In this study, the thermal hazard, exposure, and vulnerability indices were used to construct the final thermal risk index. The thermal hazard index was expressed by the inversion of Landsat 8 data, and the accuracy of the inversion results was verified by combining them with meteorological station data (Figure 5). The heat exposure index was represented by the population density data and verified using population statistics. The verified results showed that the R^2^ was 0.70. Further, the representative thermal vulnerability index of the population vulnerable to heat stress was used to calculate the spatial distribution characteristics of the overall thermal risk in the study area (Figure 9). As shown in the figure, the thermal risk index was the highest in the urban center and gradually decreased from the center to the surrounding area. In addition, the thermal risk index of the construction area on the northern side of the Hun River was significantly higher than that on the southern side because many old urban areas of Shenyang were on the northern side of the Hun River, which was often well built and populous. According to Figure 9, the thermal risk index of the building-type LCZ was higher than that of the natural-type LCZ.

In this study, the LCZ types and thermal risk indices were superimposed to analyze the thermal risk differences caused by different LCZ types (Figure 10). The results revealed that the thermal risk indices of LCZ building types were significantly higher than those of natural types (Table 4). Within the building types, LCZ 8, which included open middle high-rise buildings, had the highest average thermal risk index (0.48). The second was LCZ 3, which comprised compact middle high-rise buildings (0.46), and with decreasing building height and density, the thermal risk index gradually decreased. Among the natural-type LCZs, LCZ E (bare rock and paved areas) and LCZ F (bare soil and sandy areas) had the highest thermal risk indices, reaching 0.31 and 0.29, respectively. For other natural types, the thermal risk indices were stable between 0.20 and 0.21.

## 4. Discussion

In this study, the LCZ system was used to classify Shenyang City. This framework classifies the urban surface based on the differences in ground cover, land nature, and human activity, and has been widely used in studying the UHI effect. Many scholars have mapped LCZs at grid and community scales to analyze the differences between thermal environments [45,46,47,48]. However, while analyzing the grid scale, the consistency of multi-source data in terms of spatial resolution should be considered. When urban heterogeneity is high, resampling and other methods can easily cause loss of information. Obtaining timely statistical data on administrative boundary changes and community levels on a community scale is difficult. The General Spatial Planning of Shenyang City (2021–2035) Report advises that, while optimizing the overall spatial pattern, the spatial structure of the central urban area should be adjusted to form “one main area and three secondary areas, one riverbank, one corridor, and two axes.” Therefore, a grid of 100 m × 100 m was used to divide the LCZ in Shenyang.

Against the background of global warming, the likelihood of urban residents receiving heat threats is increasing. Presently, most studies on thermal risk are concentrated in developed countries, whereas relevant studies show that tropical areas and developing countries are more susceptible to the impact of thermal risk [49]. Particularly, the education and medical systems in urban areas attract the younger generations to cities, while the older population remains in rural areas. Additionally, as medical resources and economic conditions in rural areas lag far behind those in urban areas, the elderly are more vulnerable to the threat of heat.

Gao et al. [50] considered the number of air conditioners in constructing a final thermal risk index. Traditional thermal risk assessment usually combines several different vulnerable groups into a general thermal risk index, which commonly includes temperature, population, and socioeconomic indices. However, owing to differences in the scale of statistical data, using statistical methods for index calculations is usually necessary. For example, in China, population statistics are usually conducted according to administrative levels and are published on various government websites. Acquiring population data at a level below the county level is difficult, and conducting quantitative spatial analysis is challenging. Therefore, this study combined world population density data with a spatial resolution of 100 m and LST data and proportionally placed the elderly population in a grid to better describe the spatial distribution characteristics of heat risk. The spatial pattern of thermal risk was consistent with the surface temperature and population density layers, indicating that thermal hazard and exposure were the leading factors to an increased thermal risk index. Finally, this study analyzed the differences among the thermal risks from the perspective of LCZ because the system better reflects these differences in the underlying urban surfaces. Compared with using a single index to construct a thermal risk index, LCZ classification more completely expresses regional landscape differences. To avoid urban thermal risk, more targeted spatial information can be provided to improve the comparability of thermal risk assessments between different cities.

This study analyzed urban thermal risk from the perspective of LCZ, but some deficiencies still need further analysis and research. First, because of the failure to obtain better quality data, this study only used building height and density to divide the building-type LCZs. Higher quality building data should be used for classification, in future studies. Second, to calculate the thermal risk index, this study only used three indices, which may have certain limitations compared with the multi-index quantification performed by other scholars. This is because spatially quantifying the statistics is difficult. In future studies, more emphasis should be given to data with spatial characteristics to better calculate the spatial distribution of thermal risk.

## 5. Conclusions

This study examined the thermal risk of Shenyang from the viewpoint of LCZ by combining LST, population density, and demographic data and analyzing the differences in thermal risk among different LCZs. The main findings are as follows:

(1) In terms of spatial distribution, the LCZs of building types were primarily in the center of the study area and distributed along both sides of the Hun River, with LCZ 10 accounting for the largest proportion. Natural-type LCZs, were mainly dominated by LCZ A and LCZ C in the southwestern and northern sides of the study area, respectively.

(2) The results of the thermal risk index revealed that the thermal risk of the urban center was the highest, and it gradually decreased toward the periphery. The building-type LCZs had significantly higher thermal risk indices than the natural types, with LCZ 8 and LCZ 3 having the highest average thermal risk indices. Among the natural-type LCZs, LCZ E and LCZ F had the highest thermal risk indices.

The findings of this study can serve as a reference to mitigate the UHI effect and effectively plan urban land use.

## Figures and Tables

**Figure 1 ijerph-20-03283-f001:**
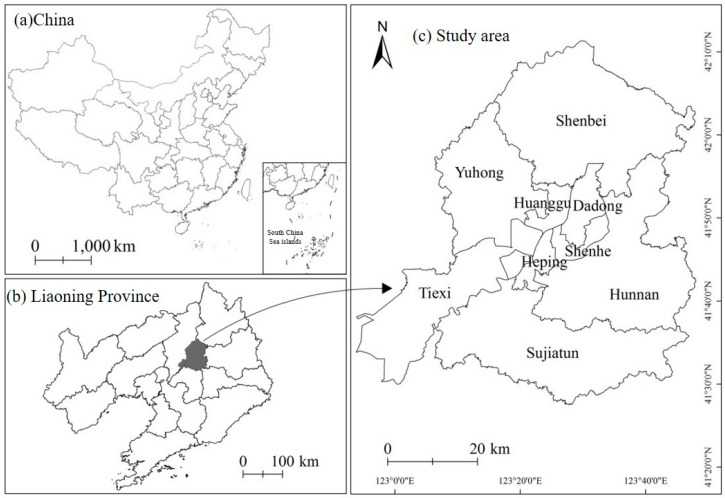
Location of study area.

**Figure 2 ijerph-20-03283-f002:**
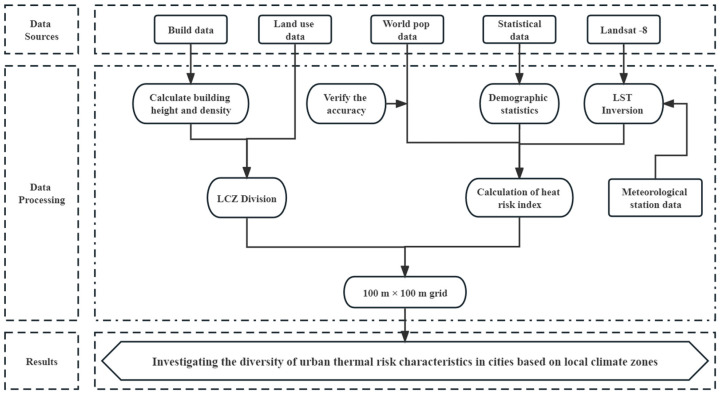
Research flowchart.

**Figure 3 ijerph-20-03283-f003:**
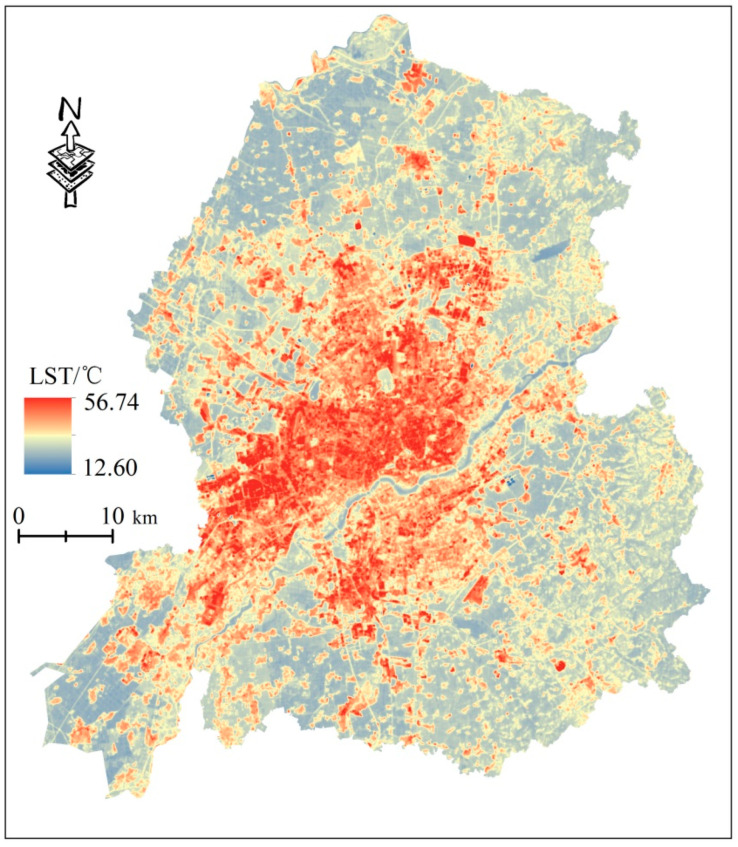
Land surface temperature inversion results.

**Figure 4 ijerph-20-03283-f004:**
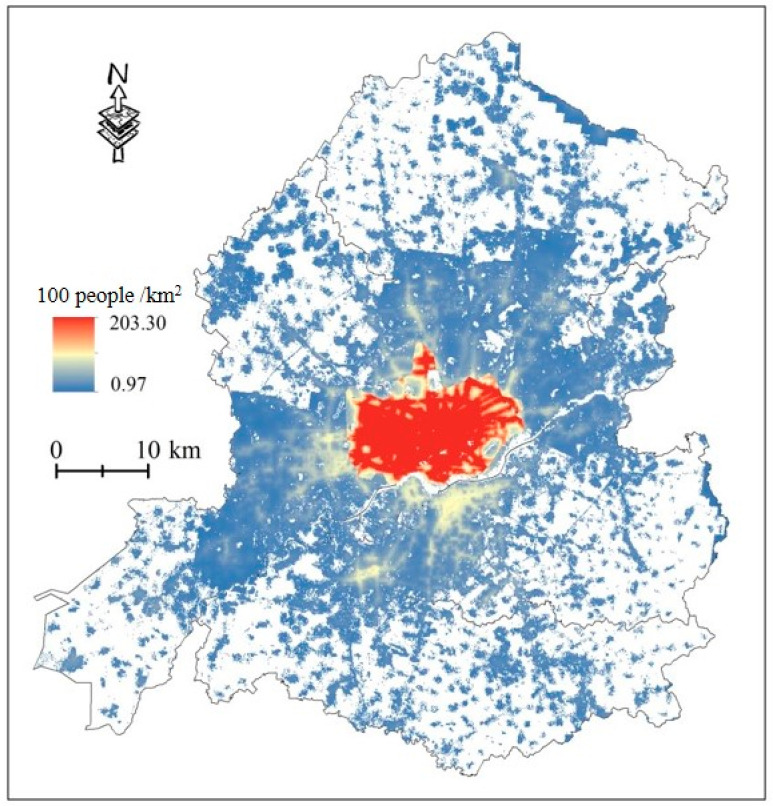
The distribution of population density in Shenyang city center.

**Figure 5 ijerph-20-03283-f005:**
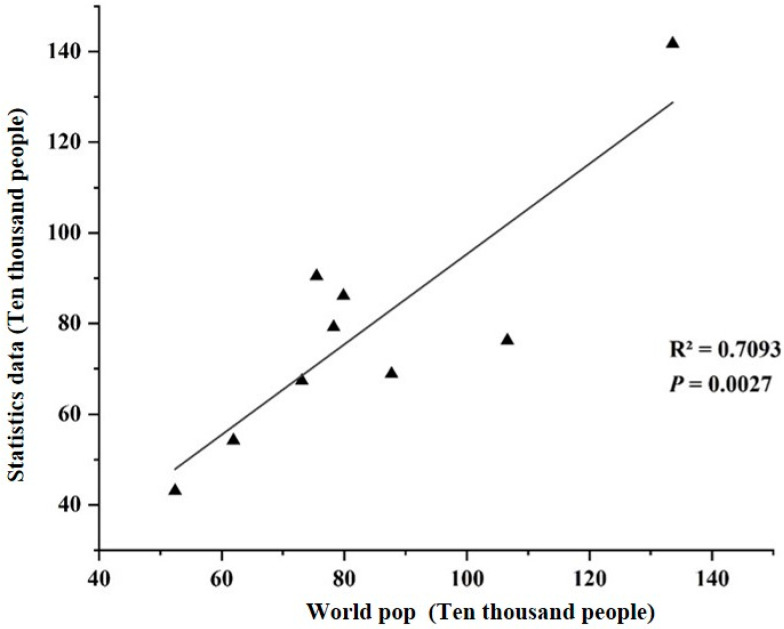
Accuracy verification of population density data.

**Figure 6 ijerph-20-03283-f006:**
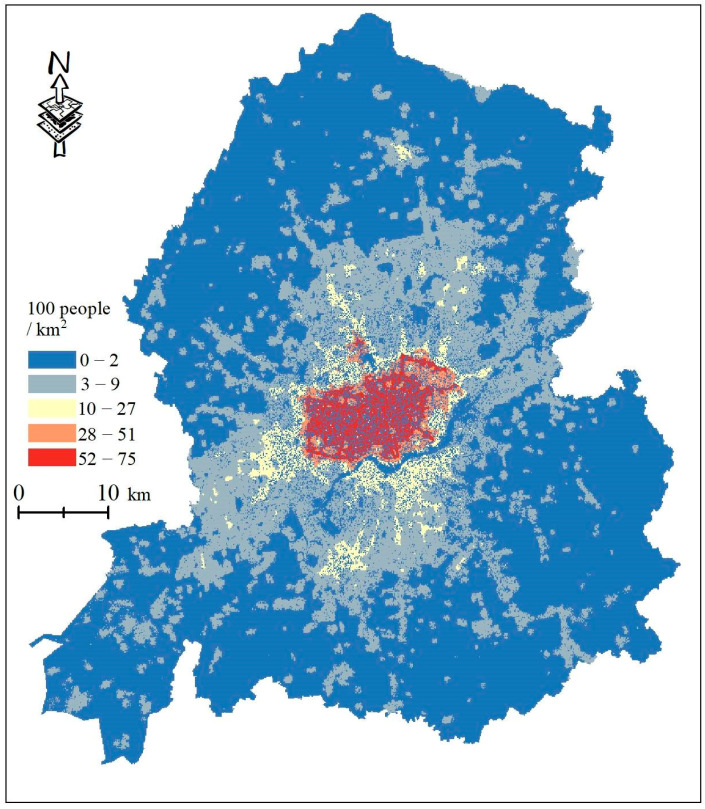
Spatial distribution of vulnerable populations.

**Figure 7 ijerph-20-03283-f007:**
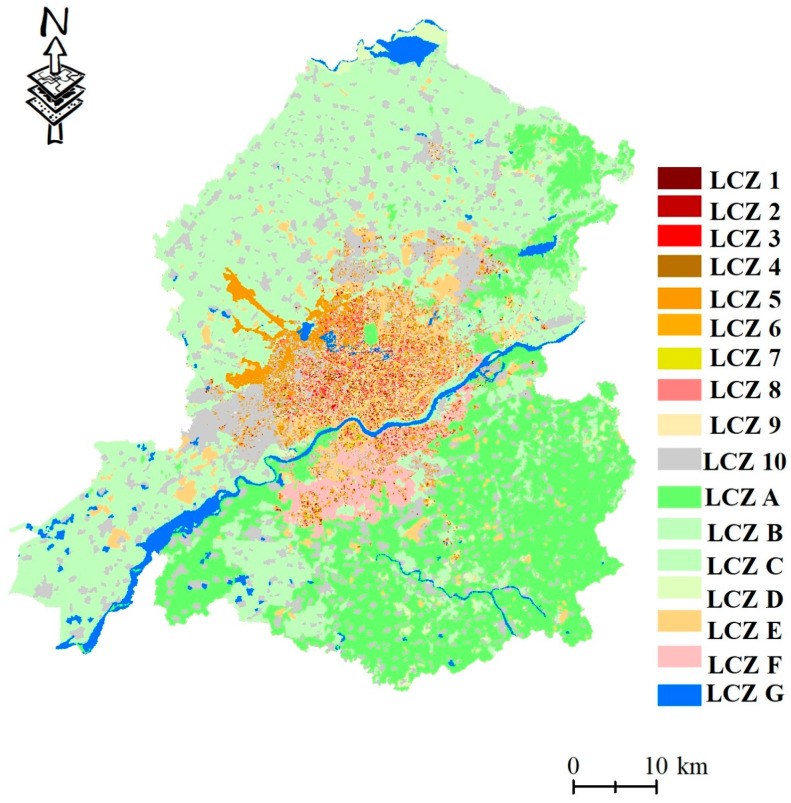
LCZ division results and the number of each type.

**Figure 8 ijerph-20-03283-f008:**
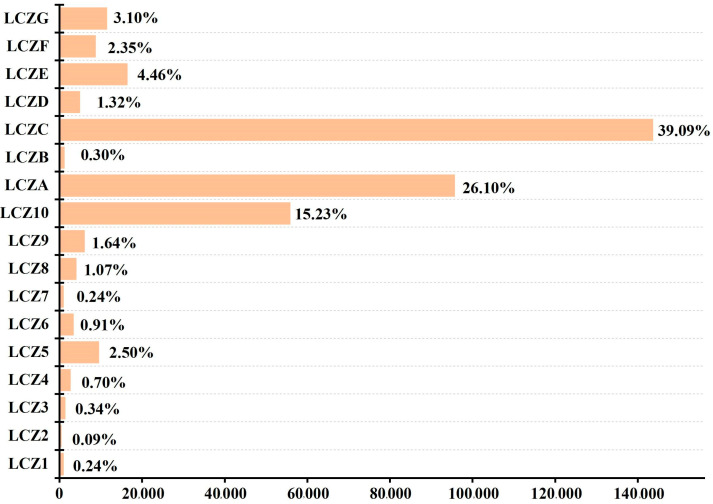
The number of each LCZ type.

**Figure 9 ijerph-20-03283-f009:**
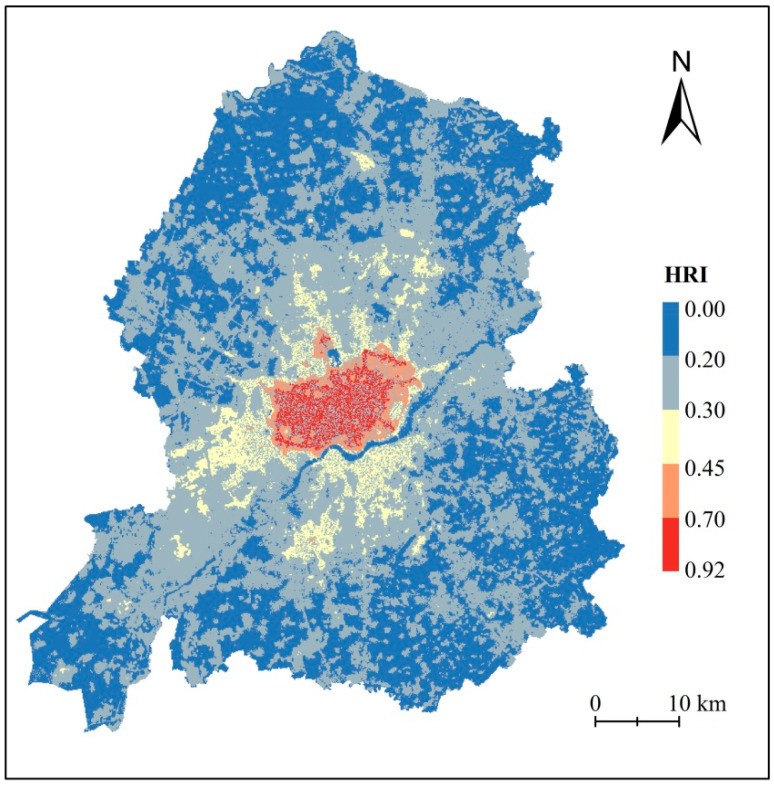
Spatial distribution of heat risk index.

**Figure 10 ijerph-20-03283-f010:**
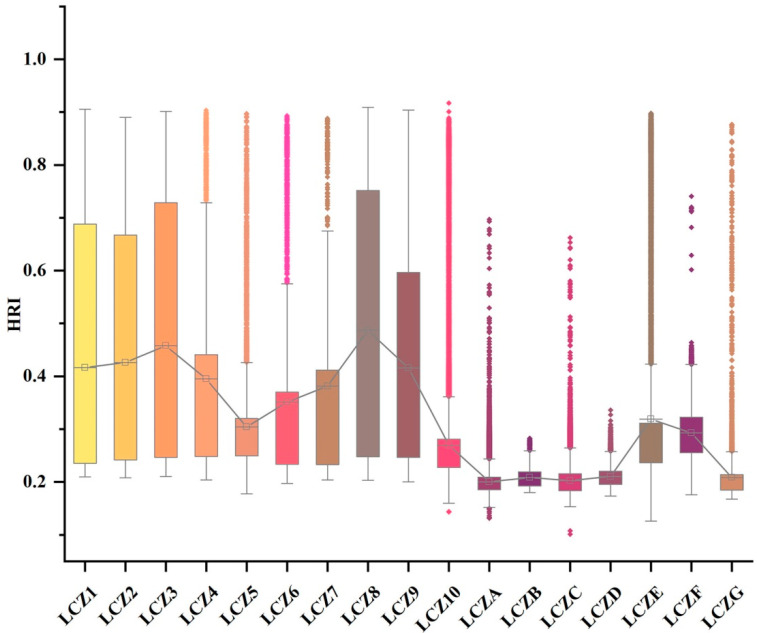
Heat risk index differences among different LCZ types.

**Table 1 ijerph-20-03283-t001:** Data sources and description.

Data Types	Time	Resolution	Sources
Landsat-8	2020.7.22	30 m	http://www.gscloud.cn/ (accessed date: 21 October 2022)
Build data	2018	-	https://map.baidu.com/ (accessed date: 21 October 2022)
Land use data	2018	30 m	http://doi.org/10.5281/zenodo.4417809 (accessed date: 26 October 2022)
WorldPop data	2020	100 m	https://www.worldpop.org/ (accessed date: 21 October 2022)
Socioeconomic and statistical data	2020	-	http://tjj.shenyang.gov.cn/ (accessed date: 24 October 2022)

**Table 2 ijerph-20-03283-t002:** Calculation of urban form parameters.

Index	Calculate Formula	Description
Building density (BD)	$\mathrm{BD}=\frac{S_{\mathrm{build}}}{S_{\mathrm{area}}}$	$S_{\mathrm{build}}$ represents the building base area in the unit grid, $S_{\mathrm{area}}$ represents the grid area
Building height (BH)	$\mathrm{BH}=\frac{\sum_{I=1\;}^{N}H_{I}}{N}$	$H_{I}$ is the height of i buildings in the grid, N is the number of all buildings in the grid

**Table 3 ijerph-20-03283-t003:** LCZ types.

Building LCZs	Explanation	Nature LCZs	Explanation
LCZ 1	Compact super-high-rise (Above 12 floors)	LCZ A	Dense trees
LCZ 2	Compact high-rise (10–12 floors)	LCZ B	Scattered trees
LCZ 3	Compact middle-high-rise (7–9 floors)	LCZ C	Bush
LCZ 4	Compact mid-rise (4–6 floors)	LCZ D	Grass
LCZ 5	Compact low-rise (1–3 floors)	LCZ E	Bare rock and paved
LCZ 6	Open super-high-rise (Above 12 floors)	LCZ F	Bare soil and sand
LCZ 7	Open high-rise (10–12 floors)	LCZ G	Water
LCZ 8	Open middle-high-rise (7–9 floors)		
LCZ 9	Open mid-rise (4–6 floors)		
LCZ 10	Open low-rise (1–3 floors)		

**Table 4 ijerph-20-03283-t004:** Statistical results of heat risk index of different LCZ types.

LCZ Types	Min	Max	Avg	Std
LCZ 1	0.21	0.91	0.42	0.24
LCZ 2	0.20	0.89	0.43	0.25
LCZ 3	0.21	0.90	0.46	0.20
LCZ 4	0.20	0.90	0.40	0.09
LCZ 5	0.18	0.89	0.30	0.18
LCZ 6	0.20	0.89	0.35	0.21
LCZ 7	0.20	0.88	0.38	0.24
LCZ 8	0.20	0.91	0.48	0.22
LCZ 9	0.20	0.90	0.41	0.02
LCZ 10	0.14	0.92	0.27	0.08
LCZ A	0.00	0.69	0.21	0.02
LCZ B	0.17	0.28	0.20	0.03
LCZ C	0.00	0.66	0.20	0.25
LCZ D	0.17	0.33	0.21	0.02
LCZ E	0.13	0.89	0.31	0.15
LCZ F	0.17	0.74	0.29	0.05
LCZ G	0.00	0.87	0.21	0.06

## Data Availability

The original datasets used in the study are included in the article. Further inquiries can be directed to the corresponding author.

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
