# Peer review of "Variations of Urban Thermal Risk with Local Climate Zones"

_ijerph, 2023, doi:10.3390/ijerph20043283_

Round 1
Reviewer 1 Report
The reviewer thanks the authors for the submission and the Editor for the invitation to review it.
In the presented manuscript, the authors examined the dependence of heat risk in cities on local climatic zones of LCZ. They based their research on downtown Shenyang and used the LCZ system to assess thermal risk. They used the most important information about LST, population density and demographic data for the research. The results presented by the authors show that the type of LCZ significantly impacts the occurrence of the thermal risk. The work deals with important and current topics.
The reviewer has reservations about the manuscript on several points:
1. Line 114. There is a lot of data here; where did it come from? It is necessary to cite the source, especially due to the multi-source data.
2. Line 120. What were the criteria for selecting and attempting urbanization to use the statement of high urbanization?
3. Figure 1. In Figure 1, the area in the upper left corner is not marked.
4. Table 1. How does a different resolution affect the final results?
5. Figure 2. Is the adopted procedure supported by the standard, previous literature reports?
6. Table 3. The table is illegible; it will be necessary to separate building LCZs from Nature LCZs.
7. Figure 7. Units and axis descriptions are missing. In addition, there is relatively poor readability. Maybe it's worth considering dividing the drawing into two and putting them one under the other?
8. Line 255-256. The discussion sentence indicates that it is difficult to compare the test results reported in the literature due to multi-source data. To what extent does the work reported in this article answer this issue?
9. Line 419 and 429. Duplicated literature.
10. Some figures are missing units. Please check the entire manuscript.
11. There is no clearly defined purpose and justification for the research. What does this research bring in the scientific context (what is the scientific element of this work)? What benefits will the use of this research bring, and in what directions?
12. The important point is to indicate the substantive differences between the submitted manuscript and the cited reference [3].
Author Response
Thanks for the reviewer’s suggestion. We have revised the paper based on the other reviewers’ comments, if you have any new suggestion or question, please contact us without hesitation.

Reviewer 2 Report
Overall, the article is well organized and is recommended to be accepted after modification. Some minor issues still need to be improved:
1. How is the world risk index mentioned in the introduction related to the thermal risk? And this citation lacks relevant theoretical support.
2. Why the years of the multi-source data used in this paper are not harmonized and whether there is temporal variability.
3. All the calculation method are supposed to be justified to be reliable to use
4. The discussion section lacks a comparative approach to those studies addressed in the literature review section.
Author Response

(The authors gave the same response as above.)

Round 2
Reviewer 1 Report
The author has responded to previous review comments, which I accept.
There is only one thing left to consider. The manuscript contains much information from the literature [3]. To avoid accusations of plagiarism, I suggest referring to the research methodology from this scientific article and not describing it again in the current manuscript.
Author Response
Many thanks to the reviewer for revising the article again. The author has modified the existing problems. If you fail to accurately answer your questions, please contact us immediately.